# WaveMix: Resource-efficient Neural Network for Image Analysis

## Abstract

We propose a novel neural architecture for computer vision – WaveMix – that is resource-efficient and yet generalizable and scalable. While using fewer trainable parameters, GPU RAM, and computations, WaveMix networks achieve comparable or better accuracy than the state-of-the-art convolutional neural networks, vision transformers, and token mixers for several tasks. This efficiency can translate to savings in time, cost, and energy. To achieve these gains we used multi-level two-dimensional discrete wavelet transform (2D-DWT) in WaveMix blocks, which has the following advantages: (1) It reorganizes spatial information based on three strong image priors – scale-invariance, shift-invariance, and sparseness of edges – (2) in a lossless manner without adding parameters, (3) while also reducing the spatial sizes of feature maps, which reduces the memory and time required for forward and backward passes, and (4) expanding the receptive field faster than convolutions do. The whole architecture is a stack of self-similar and resolution-preserving WaveMix blocks, which allows architectural flexibility for various tasks and levels of resource availability. WaveMix establishes new benchmarks for segmentation on Cityscapes; and for classification on Galaxy 10 DECals, Places-365, five EMNIST datasets, and iNAT-mini and performs competitively on other benchmarks.

## 1 Introduction

Natural images have a number of priors that are not comprehensively exploited in any single type of neural network architecture. For instance, (1) convolutional neural networks (CNNs) only model shift-invariance using convolutional design elements (Lecun et al., 1998; Krizhevsky et al., 2012; Simonyan & Zisserman, 2014; Szegedy et al., 2014; He et al., 2015; Howard et al., 2017; Hu et al., 2017; Huang et al., 2016), (2) vision transformers (ViT) model long-range dependencies using self-attention (Zhao et al., 2020; Dosovitskiy et al., 2021), and (3) token mixers also model the long-range dependencies but without the quadratic complexity of self-attention (Tolstikhin et al., 2021; Guibas et al., 2021; Trockman & Kolter, 2022). However, none of these architectures exploit other image priors, such as scale-invariance and spatial sparseness of edges. Self-attention amplifies low-frequency components while convolutions amplify high frequency components in images (Park & Kim, 2022). Additionally, transformers and token-mixers cannot easily be adapted for per-pixel tasks, such as segmentation, without significant architectural changes (Trockman & Kolter, 2022).

Table 1: A sample of WaveMix generalization performance and parameters compared with previous state-of-the-art (SOTA). See text for results on additional datasets.

| Task | Metric | Datasets | | | Previous SOTA | | | WaveMix | |
|---|---|---|---|---|---|---|---|---|---|
| | | Pre-train | Train | Test | Ref. | Perf. | Param. | Perf. | Param. |
| Segmentation | mIoU (SS) | ImageNet-1k | Cityscapes | Cityscapes val. | (Xie et al., 2021) | 82.40 | 85 M | **82.70** | **63 M** |
| Classification | Accuracy | - | Places-365 | Places-365 val. | (Wang et al., 2020b) | 56.32 | 31 M | **56.45** | **28 M** |
| | | - | EMNIST Letters | EMNIST Letters | (Kabir et al., 2020) | 95.88 | **4 M** | **95.96** | **4 M** |
| | | ImageNet-1k | Galaxy 10 DECals | Galaxy 10 DECals | (Dagli, 2023) | 94.86 | 272 M | **95.42** | **28 M** |

We solve the challenges of architectural flexibility and comprehensive exploitation of natural image priors by proposing a novel neural architecture for computer vision called *WaveMix*, which generalizes better with fewer resources for a variety of tasks, as shown in Table 1.

At the heart of WaveMix are three design elements – a stack of self-similar WaveMix blocks, a *multi-level* two-dimensional discrete wavelet transform (2D-DWT) within each block, and spatial resolution contraction followed by expansion back to the original size within a block. We use the multi-level 2D-DWT for resource-efficient token-mixing and to exponentially increase the receptive field. Additionally, 2D-DWT transform partitions the image features into different frequencies and processes them separately, allowing the model to focus equally on both low-frequency and high-frequency component. These design elements are our key contributions, along with adaptations, extensive experimentation, and analysis to demonstrate the versatility, utility, and parsimony of the WaveMix design.

We relate WaveMix to previous works in Secion 2, where we delve further into the image priors modeled by various classes of neural architectures for vision, and the use of wavelet transform. Our key innovations – the WaveMix blocks, use of multi-level 2D-DWT in each block, channel mixing, and the preservation of feature tensor dimension – and their rationale are explained in Section 3. In Section 4 we show comprehensive experiments and results that compare WaveMix with several other state-of-the-art (SOTA) models for multiple vision tasks and show that WaveMix models can match or outperform much larger models in generalization. In addition to generalization, we provide evidence of scalability (adding WaveMix blocks or channels to improve generalization), and parsimony (smaller number of parameters, GPU RAM requirement, and inference time) of the WaveMix architecture.

## 2 Background and Related Works

**Statistical properties of natural images** that have been well-studied include shift-invariance, scale-invariance, high spatial auto-correlation and preponderance of certain colors, as well as spatial sparseness of edges (Field, 1993; Ruderman, 1994; Lee, 1996; Párraga et al., 2002). Shift-invariance, which is a form of stationarity of signals, arises due to the assumption of uniform distribution over the location of objects and visual features. Scale-invariance arises from the possibility of an object being at any distance from the camera, and the hierarchical nature of groups of objects, objects, their parts, and parts of parts, etc (Zoran & Weiss, 2009). High auto-correlation arises from each object being relatively more homogeneous within as compared to the other objects in the scene. The finite spatial extents of objects lead to spatially sparse occlusion boundaries that manifest as edges. Natural selection has facilitated survival of camouflaged species of lifeforms (which are popular salient objects in images) that have edge-like patterns on their outer surfaces.

**Two-dimensional discrete wavelet transform (2D-DWT)** has been extensively researched to exploit various properties of images for multiple applications, including denoising (Ruikar & Doye, 2010), super-resolution (Guo et al., 2017), recognition (Mahmood et al., 2018), and compression (Lewis & Knowles, 1992). Features extracted using wavelet transforms have also been used extensively with machine learning models (Mowlaei et al., 2002), such as support vector machines and neural networks (Ranaware & Deshpande, 2016), especially for image classification (Nayak et al., 2016). Some instances of integration of wavelets in neural architectures include the following. ScatNet architecture cascades wavelet transform layers with non-linear modulus and average pooling to extract translation-invariant features that are robust to deformations and preserve high-frequency information for image classification (Bruna & Mallat, 2013). WaveCNets replaces max-pooling, strided-convolution, and average-pooling of CNNs with 2D-DWT for noise-robust image classification (Li et al., 2020). Multi-level wavelet CNN (MWCNN) has been used for image restoration for a better trade-off between receptive field size and computational efficiency (Liu et al., 2018). Wavelet transform has also been combined with a fully convolutional neural network for image super-resolution (Kumar et al., 2017). Pooling using wavelets have been shown to perform comparably with other pooling techniques (Williams & Li, 2018). WaveSNet uses DWT to downsample and inverse-DWT to upsample feature maps for segmentation (Li & Shen, 2020). Recently, wavelet transform has been used with self-attention in transformers (Patro & Agneeswaran, 2023).

What makes DWT an attractive tool for analysis of natural signals are its multi-resolution properties and treatment of spatio-temporally sparse discontinuities (edges). A 1D-DWT splits an input 1-D signal $\mathbf{x}$ of

length $H$ into two sub-bands roughly of length $H/2$ each (Daubechies, 1990). The first one is called the approximate sub-band $w_a(\mathbf{x})$, which can be thought of as the lower-resolution version of the original signal. In its simplest form as a Haar wavelet, $w_a(\mathbf{x})$ is proportional to the sum of two consecutive samples of the input, followed by downsampling by a factor of 2. The second one is called the detail sub-band $w_d(\mathbf{x})$, which captures the information lost by the approximation sub-band. In Haar wavelet, it is proportional to the difference of two consecutive samples, followed by a downsampling operation. That is, both sub-bands are convolution with a $2 \times 1$ filter with stride 2 and can be thought of as *local* low and high-frequency components, respectively, that together are orthogonal and compact the signal energy (Mallat, 1989).

A 2D-DWT is the application of 1D-DWT on the rows of a 2-D signal, followed by a column-wise 1D-DWT of the resultant rows. A 2D-DWT has one approximation $w_{aa}$ and three detail sub-bands $w_{ad}, w_{da}, w_{dd}$ (Kumar & Sethi, 2018). The approximation sub-band can be further decomposed using a repeated application of the wavelet transform, and the resultant decomposition is known as level 2 transform. All sub-bands model shift-invariance, the detail sub-bands model spatial derivatives (e.g., edges), and the levels model spatial scale (Daubechies, 1990).

**CNN** generalization accuracy for various vision tasks has improved over the last decade due to architectural innovations that ease gradient flow to deeper layers (He et al., 2015; Szegedy et al., 2014),or reduce parameters per layer by restricting convolutional kernel size or dimension (Simonyan & Zisserman, 2014; Chollet, 2016), or multi-scale feature aggregation using multiple branches (Chen et al., 2019). Attention layers for space or channel seem to improve the performance of CNNs to some extent (Chen et al., 2017), but these have not been widely adopted for vision tasks due to a lack of advantage in deeper CNN architectures.

**Vision transformers (ViTs)**, inspired by the success of transfomers on NLP tasks, use a one-dimensional sequence of tokens corresponding to image patches (sub-images, tiles) (Dosovitskiy et al., 2021). Although, ViTs have proven to be scalable models that generalize better than CNNs for image classification, the destruction of potentially useful 2-D structure during patch tokenization is countered by the use of very large datasets and number of parameters to re-learn class-specific spatial associations in the destroyed direction (say, vertical) in an unintuitive way. Training these models requires access to multiple GPUs with large RAMs. While there is some long-range association between visual features to be learned for image-based reasoning, the use of quadratic self-attention may be an overkill, unlike its clear advantage in natural language processing.

**Hybrid models** aim to reduce the computational requirements of vision transformers by incorporating image-specific inductive biases in the form of additional architectural elements including distillation (Touvron et al., 2020), convolutional embeddings (Jeevan & Sethi, 2022; Hassani et al., 2021), convolutional tokens (Wu et al., 2021), and encoding overlapping patches (Yuan et al., 2021). The quadratic complexity with respect to the sequence length (# of tokens) of transformers has also led to the search for other linear approximations of self-attention for efficiently mixing tokens (Jeevan & Sethi, 2022). However, these models suffer from similar inflexibility as the ViTs.

**Token mixers** that replace the self-attention in transformers with fixed token mixing mechanisms, such as the Fourier transform (Guibas et al., 2022; Rao et al., 2021), achieves comparable generalization with lower computational requirements on NLP tasks (Lee-Thorp et al., 2021). Other token-mixing architectures (Yu et al., 2021) have also been proposed that use standard neural components, such as convolutional layers and multi-layer perceptrons (MLPs, i.e. $1 \times 1$ convolutional filters) for mixing visual tokens. MLP-mixer uses two MLP layers applied first to an image patch sequence and then to the channel dimension to mix tokens (Tolstikhin et al., 2021). ConvMixer uses standard convolutions along spatial dimensions and depth-wise convolutions across channels to mix tokens with fewer computations than attention mechanisms (Trockman & Kolter, 2022). PoolFormer uses pooling as a token-mixing operation (Yu et al., 2021). These token mixing models perform efficiently compared to transformers without compromising generalization. However, these models do not use image priors well.

Our work combines ideas from CNNs, token mixers, and wavelet transforms with innovative architectural elements to use image priors in a computationally and parametrically more efficient manner.

# 3   WaveMix Architectural Framework

Figure 1: WaveMix architecture for (a) image classification and (b) semantic segmentation, along with (c) details of the WaveMix block

Due to the properties of wavelet transform mentioned in the previous section – shift-invariance, multi-resolution analysis, edge-detection, local operations, and energy compaction – we propose using it in neural network architectures for computer vision as token mixers, feature reorganizers, and spatial compactors with fixed weights as described below.

## 3.1   Overall architecture

As shown in Figure 1 (a) and (b), the macro-level idea behind the proposed framework is to stack $N$ (a hyperparameter) similar WaveMix blocks that are fully convolutional (by design) in both spatial dimensions *and* maintain the spatial resolution of the feature maps across the blocks. While some CNNs are fully convolutional, vision transformer architectures tend to maintain the sequence lengths (which is a flattened version of the spatial resolution). Combining these two elements gives WaveMix architectural flexibility to easily adapt to various image dimensions and tasks. For instance, while pixel maps for semantic segmentation or image enhancement are a natural outcome of this framework, adding an optional MLP layer for channel

mixing followed by global average pooling and softmax are enough to adapt it for classification of images of various sizes. That is, WaveMix processes image as a 2D grid and not as sequence of pixels/patches by unrolling the image as done in transformer models. For additional flexibility, we pass the input image $x' \in \mathbb{R}^{H \times W \times 3}$ through an initial convolutional layer which increases the number of channels to embedding dimension $C$ (i.e., $x \in \mathbb{R}^{H \times W \times C}$, where $C$ is a hyperparameter) before passing it to WaveMix blocks. Optionally, the initial convolutional layer can also be used to reduce the input resolution either using strided or patchifying convolutions (Trockman & Kolter, 2022; Liu et al., 2022).

## 3.2 WaveMix block

As shown in Figure 1 (c), the design of the WaveMix block is such it that does not collapse the spatial resolution of the feature maps, unlike CNN blocks that use pooling operations (He et al., 2015). And yet, it reduces the number of computations required by reducing the spatial dimensions of the feature maps using 2D-DWT, which translates to a reduction in GPU RAM, training time, and inference time. However, unlike pooling or strided convolutions, a 2D-DWT is lossless as it expands the number of channels by the same factor by which it reduces spatial resolution. Furthermore, it has additional energy compaction (sparsification) properties that are not offered by random filters or Fourier basis.

Denoting input and output tensors of the WaveMix block by $\mathbf{x}_{in}$ and $\mathbf{x}_{out}$, respectively; level of the wavelet transform by $l \in \{1...L\}$, the four wavelet filters along with their downsampling operations at each level by $w_{aa}^l, w_{ad}^l, w_{da}^l, w_{dd}^l$ ($a$ for approximation, $d$ for detail); convolution, multi-layer perceptron (MLP), transposed convolution (upconvolution), and batch normalization operations by $c$, $m$, $t$, and $b$, respectively; and their respective trainable parameter sets by $\xi$, $\theta_l$, $\phi_l$, and $\gamma_l$, respectively; concatenation along the channel dimension by $\oplus$, and point-wise addition by $+$, the operations inside a WaveMix block can be expressed using the following equations:

$$\mathbf{x}_0 = c(\mathbf{x}_{in}, \xi); \qquad \mathbf{x}_{in} \in \mathbb{R}^{H \times W \times C}, \quad \mathbf{x}_0 \in \mathbb{R}^{H \times W \times C/4} \tag{1}$$

$$\mathbf{x}_l = [w_{aa}^l(\mathbf{x}_0) \oplus w_{ad}^l(\mathbf{x}_0) \oplus w_{da}^l(\mathbf{x}_0) \oplus w_{dd}^l(\mathbf{x}_0)]; \qquad \mathbf{x}_l \in \mathbb{R}^{H/2^l \times W/2^l \times 4C/4}, l \in \{1...L\} \tag{2}$$

$$\hat{\mathbf{x}}_l = [\mathbf{x}_l \oplus \tilde{\mathbf{x}}_{l+1}], \qquad \hat{\mathbf{x}}_L = \mathbf{x}_L; \quad l \in \{1...L-1\} \tag{3}$$

$$\tilde{\mathbf{x}}_l = b(t(m(\hat{\mathbf{x}}_l, \theta_l), \phi_l), \gamma_l); \quad \tilde{\mathbf{x}}_l \in \mathbb{R}^{H/2^{l-1} \times W/2^{l-1} \times C_l} \forall l > 1 \ \ C_l = C/2, \ \ C_1 = C, \ \ l \in \{1...L\} \tag{4}$$

$$\mathbf{x}_{out} = \tilde{\mathbf{x}}_1 + \mathbf{x}_{in}; \qquad \mathbf{x}_{out} \in \mathbb{R}^{H \times W \times C} \tag{5}$$

The WaveMix block extracts learnable and space-invariant features using a convolutional layer, followed by spatial token-mixing and downsampling for scale-invariant feature extraction using multi-level 2D-DWT (Lütkebohle, 2018), followed by channel-mixing using a learnable MLP (1×1 conv) layer, followed by restoring spatial resolution of the feature map using a transposed-convolutional layer. The use of trainable convolutions *before* the wavelet transform is a key design aspect of our architecture as it allows the extraction of only those feature maps that are suitable for the chosen wavelet basis functions. The convolutional layer $c$ decreases the embedding dimension $C$ by a factor of four so that the concatenated output $\mathbf{x}_l$ after each level of 2D-DWT has the same number of channels as the input $\mathbf{x}_{in}$ (Eq. 1 and Eq. 2). That is, since 2D-DWT is a lossless transform, it expands the number of channels by the same factor (using concatenation) by which it reduces the spatial resolution by computing an approximation sub-band (low-resolution approximation) and three detail sub-bands (spatial derivatives) (Daubechies, 1990) for each input channel (Eq. 2). Use of this image-appropriate and lossless downsampling using 2D-DWT allows WaveMix to use fewer layers and parameters. The four outputs from each level are concatenated together to form $\mathbf{x}_l$ as shown in Eq. 2.

We decide the number of levels of wavelet decomposition $L$ based on the image size. Each level $l$ reduces the output size by a factor of $2\times 2$. Our experiments show that using three or four levels of 2D-DWT are sufficient to ensure token-mixing over long spatial distances. For instance, a 3-level DWT decomposition of an image of size $128\times 128$ simultaneously creates feature maps of size $64\times 64$, $32\times 32$ and $16\times 16$, which are sufficient for spatial token mixing. Using more levels of DWT increases the receptive field exponentially, but it also amplifies noise in the model (Daubechies, 1990), which degrades the performance.

The concatenated output from each level of DWT $\mathbf{x}_l$ is concatenated with the output from the next higher level path $\tilde{\mathbf{x}}_{l+1}$ as shown in Eq. 3. The output $\hat{\mathbf{x}}_l$ is then passed to an MLP layer $m$, which has two $1 \times 1$ convolutional layers with an inverse bottleneck design (multiplication factor $> 1$) separated by a GELU non-linearity. After this, the feature map resolution is doubled using transposed-convolutions $t$ followed by batch normalization $b$ (Eq. 4). We do not use transposed-convolutions in each DWT level path to directly resize image back to the original input resolution because doing so will require larger filter sizes, which will lead to exponential increase in parameters.

The number of channels in output of higher level DWT paths are halved using transposed convolutional layer $t$ before using batch norm $b$ (Eq. 4). The output $\tilde{\mathbf{x}}_l$ is concatenated with DWT output $\mathbf{x}_{l-1}$ of the previous level (Eq. 3). Hence, there is an accumulation of features from the higher levels of DWT to lower levels and the final output after all the token-mixing and feature extraction is obtained from the level 1 path as $\tilde{\mathbf{x}}_1$. A residual connection is used to ease the flow of the gradient (He et al., 2015) (Eq. 5).

The feed-forward (MLP) sub-layers immediately following DWT have access to the outputs at the corresponding level and those from higher levels (after transposed-convolution) to learn both local and global features. As the information from higher levels propagate to lower levels, mixing of local and global information enables the model to learn efficiently with fewer parameters and layers. *This multi-resolution mixing of information also leads to an exponential expansion of receptive field within each block, much faster compared to CNN layers.* Just using a $2 \times 2$ kernel, DWT is able to reach $2^L \times 2^L$ pixels within each WaveMix block (Appendix).

Among the different types of mother wavelets available, we used the Haar wavelet (a special case of the Daubechies wavelet (Daubechies, 1990), also known as Db1), which is frequently used due to its simplicity and faster computation. Haar wavelet is both orthogonal and symmetric in nature, and has been extensively used to extract basic structural information from images (Porwik & Lisowska, 2004). For even-sized images, it reduces the dimensions exactly by a factor of 2, which simplifies the designing of the subsequent layers.

## 4 Experiments and Results

We compared WaveMix models with various CNN, transformer, and token-mixing models for semantic segmentation and image classification. Ablation studies were conducted to assess the effect of the hyperparameters and the importance of each component and its placement in the WaveMix block.

### 4.1 Tasks, datasets, and models compared

For semantic segmentation, we used the Cityscapes (Cordts et al., 2016) dataset. The official training dataset itself was split into training and validation sets. Results of the other models compared were directly taken from their original papers as cited in Table 2. Since ConvMixer (Trockman & Kolter, 2022) was never used for semantic segmentation, the classification head of ConvMixer was replaced with a segmentation head similar to WaveMix for segmentation experiments. Mean intersection over union (mIoU) on the Cityscapes official validation dataset was compared as a generalization metric among the models. All results are computed with single scale (SS) inference. Inference throughput on a single GPU was reported in frames/sec (FPS).

For classification, we used multiple types of publicly available datasets, including CIFAR-10, CIFAR-100 (Krizhevsky, 2009), EMNIST (Cohen et al., 2017), Fashion MNIST (Xiao et al., 2017), SVHN (Netzer et al., 2011), Tiny ImageNet (Le & Yang, 2015), ImageNet-1K (Deng et al., 2009), Places-365 (Zhou et al., 2017), and iNaturalist2021-10k (iNAT-2021-mini) (Horn et al., 2021). Top-1 accuracy on the test set of best of three runs with random initialization is reported as a generalization metric based on prevailing proto-

Table 2: Results for semantic segmentation on Cityscapes validation set show SOTA mIoU (single scale) by WaveMix without compromising inference throughput (FPS). All models use ImageNet-1k pretrained weights unless otherwise stated. Not pre-training on ImageNet-1k is denoted by *.

| ARCHITECTURE | mIoU | # PARAM. | FPS |
|---|---|---|---|
| DeepLabV3+ (Chen et al., 2018) | 80.9 | **63 M** | - |
| Axial-DeepLab (Wang et al., 2020a) | 81.1 | - | - |
| Seg-L-Mask/16 (Strudel et al., 2021) | 81.3 | 307 M | - |
| FAN-L-Hybrid (Zhou et al., 2022) | 82.3 | 77 M | - |
| SegFormer-M5 (Xie et al., 2021) | 82.4 | 85 M | 3 |
| WaveMix-256/16 (4 level)* | 78.6 | **63 M** | 7 |
| WaveMix-256/16 (4 level) | **82.7** | **63 M** | 7 |

cols (Hassani et al., 2021). For ImageNet-1k, Places-365 and iNAT-mini datasets, we have reported Top-1 accuracy in validation set. GPU RAM usage is reported in the supplementary materials.

To evaluate the performance of WaveMix on domain specific datasets where data availability is low, we choose the task of galaxy morphology classification on Galaxy 10 DECals dataset (Leung & Bovy, 2018). There is lack of efficient methods that can extract information from astronomical surveys to classify galaxies and creating large amount of annotated data is expensive. Galaxy 10 DECals dataset contains 17,736 images of $256 \times 256$ resolution distributed in 10 classes. The dataset is also imbalanced, with some classes having more images than others.

For WaveMix model notation, we use the format *Model Name -Embedding Dimension/ no. of blocks* and mention the number of levels of DWT in brackets. We call the WaveMix model which uses only one level of 2D-DWT as *WaveMix-Lite* and it has been shown to perform well in small datasets with low resolution images. For other models, we use the same notation that is used in their papers.

### 4.2 Implementation details

We adjusted the stride (or patch size) in the initial convolutional layers in all WaveMix models that handled high-resolution images to ensure that resolution of feature maps before it reached WaveMix blocks was always less than 256 for classification and 1024 for segmentation. We only used strided convolutions in the initial convolution layers for segmentation and in classification of low-resolution image (less than $128 \times 128$) datasets. For classification of datasets with larger image resolution, patchifying convolutions were used.

For Cityscapes, we used the full-resolution $1024 \times 2048$ images for training and inference. Images were resized to $256 \times 256$ for Places-365 and $224 \times 224$ for iNAT-mini and ImageNet-1k datasets. Only horizontal flip was used as data augmentation for semantic segmentation. No data augmentations were used for classification unless mentioned other-wise.

*No pre-training was performed on any of the WaveMix models used for classification.* For classification on ImageNet-1K, we used the models available in Timm (PyTorch Image Models) library (Wightman, 2019) without using pre-trained weights and used the default Timm training script (Wightman, 2019) without augmentations. Due to limited computational resources (which actually inspired looking for an alternative neural framework), the *maximum* number of training epochs was set to 300. All experiments were done with a single 80 GB Nvidia A100 GPU. For all experiments other than ImageNet-1k classification, we used AdamW optimizer ($\alpha = 0.001, \beta_1 = 0.9, \beta_2 = 0.999, \epsilon = 10^{-8}$) with a weight decay of 0.01 during initial epochs and then used SGD with learning rate of 0.001 and momentum $= 0.9$ during the final 50 epochs (Keskar & Socher, 2017; Jeevan & sethi, 2022).

Cross-entropy loss was used for image classification and pixel-wise focal loss was used for semantic segmentation. A batch-size of 5 was used for all segmentation experiments with full resolution Cityscapes dataset.

Batch-size between 256-512 was used for classification task. We used automatic mixed precision in PyTorch during training.

## 4.3 Semantic segmentation

table 1 shows that WaveMix is the current SOTA for Cityscapes dataset in terms of single-scale inference mIoU among models pre-trained using only ImageNet-1k dataset. *Higher mIoU reported by other models (Xie et al., 2021) belong to multi-scale inference.* Performance of WaveMix on Cityscapes validation set along with the reported results of other models are shown in Table 2. Even without ImageNet-1k pre-training, WaveMix performs on par with the other models which use encoders pre-trained on ImageNet-1k. After using ImageNet pre-trained weights, WaveMix outperforms all the other CNN and transformer-based models. WaveMix is more than $2\times$ faster than previous SOTA model, SegFromer (Xie et al., 2021), in inference even on a single GPU. Further improvement may be obtained by performing multi-scale inference and pre-training on ImageNet-22k dataset.

The versatility of WaveMix is such that it can be directly used for semantic segmentation by replacing the output layer with two transposed convolution layers and a per-pixel softmax layer to generate the segmentation maps. On the other hand, architectural changes – such as encoder-decoder and skip connections (Ronneberger et al., 2015) – are required for base CNNs and transformers (Xie et al., 2021), to perform segmentation.

The lower mIoU (75.78) obtained by replacing the classification head of ConvMixer (Trockman & Kolter, 2022) with segmentation head (similar to WaveMix) shows that other token-mixing architectures, which work well for classification, may not be able to translate that performance to segmentation without significant architectural modifications. This shows the versatility of our WaveMix model. Also, the GPU consumption of WaveMix is much lower than other models.

Transposed convolution operation is the major contributor ($> 75\%$) for parameters in WaveMix architecture. We can further lower its number of parameters by replacing the transposed convolution in each block by using the parameter-free up-scaling operations such as bi-linear up-sampling and pixel-shuffle. See supplementary materials for details.

## 4.4 Image classification

Table 3 shows the performance of WaveMix for image classification on multiple datasets with image sizes ranging from $28 \times 28$ to $256 \times 256$. WaveMix achieved state-of-the-art (SOTA) accuracy of 56.45% on Places-365 standard (365 classes) validation set ($256 \times 256$) among the models that were not pre-trained on larger datasets, such as (Wang et al., 2020b). WaveMix achieves 78.78% on CIFAR-100 dataset among models that do not use pre-trained weights or extra training data. It has also achieved high accuracy on iNAT-2021-mini validation set ($224 \times 224$) which has 10,000 classes (no previous SOTA on iNAT-2021 mini has been reported). We also see from Table 3 that WaveMix models establishes a new state-of-the-art on the smaller EMNIST datasets ($28 \times 28$) by outperforming the previous best results (Kabir et al., 2020; Pad et al., 2020; Gesmundo & Dean, 2022) for Balanced, Letters, Digits, Byclass and Bymerge subsets within EMNIST (Cohen et al., 2017) without using any data augmentations. WaveMix with 28 M parameters outperforms the previous SOTA on Galaxy 10 DECals dataset, Astroformer (Dagli, 2023) with 272 M parameters.

The superior performance of WaveMix in the Galaxy Morphology Classification task highlights a critical insight: traditional convolutional models, which are biased towards texture detection(Geirhos et al., 2022), tend to struggle when it comes to discerning shapes from textured images. This limitation arises from their inherent bias towards texture over shape. In contrast, WaveMix, while still a fully convolutional architecture, integrates 2D discrete wavelet transforms (2D-DWT), allowing it to create sub-bands across different frequencies. This enables the model to separately process high and low frequencies, with the latter containing vital shape information. By effectively leveraging these lower frequencies, WaveMix demonstrates a superior ability to understand and classify the shapes of galaxies. This finding suggests that for tasks requiring a focus on shape learning rather than texture detection—an area where conventional convolutional models often fall short—WaveMix offers a significant advantage.

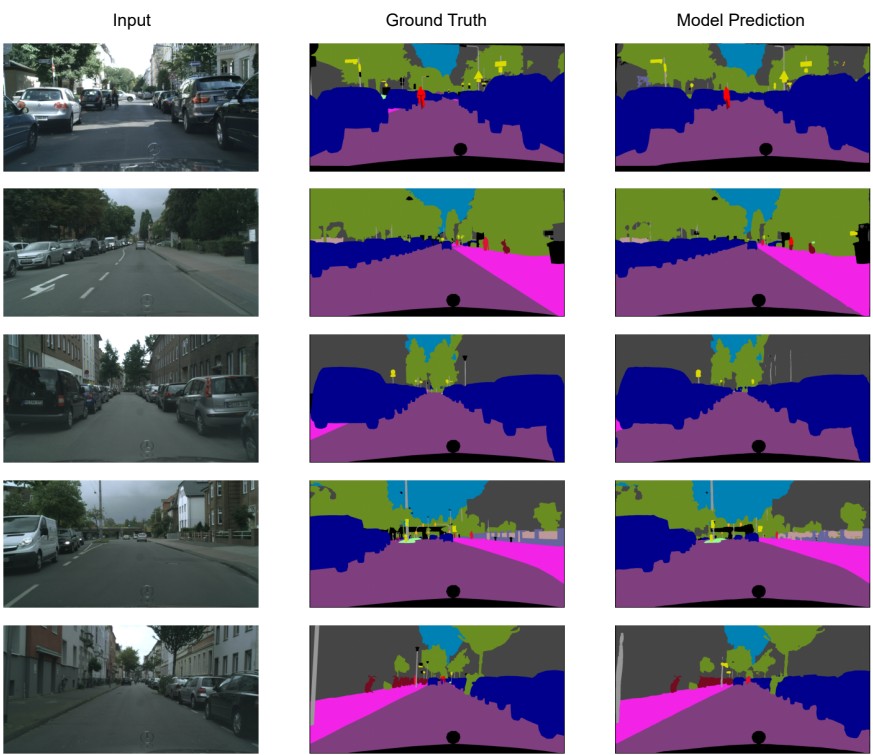

Figure 2: A sample of semantic segmentation results on Cityscapes dataset by WaveMix for qualitative assessment.

Table 3: WaveMix outperforms all previous models for image classification and achieves state-of-the-art (SOTA) results on Galaxy 10 DECals, all five EMNIST datasets, Places-365 validation set (365 classes) and INAT-mini validation set (10,000 classes). See suplementary materials for architectural details.

| DATASETS | WAVEMIX ACCURACY (%) | PREVIOUS SOTA (%) | REF. |
|---|---|---|---|
| EMNIST Byclass | **88.43** | 88.12 | (Cohen et al., 2017) |
| EMNIST Bymerge | **91.80** | 91.79 | (Cohen et al., 2017) |
| EMNIST Letters | **95.96** | 95.88 | (Kabir et al., 2020) |
| EMNIST Digits | **99.82** | **99.82** | (Gesmundo & Dean, 2022) |
| EMNIST Balanced | **91.06** | 91.05 | (Kabir et al., 2020) |
| Places-365 (val set) | **56.45** | 56.32 | (Wang et al., 2020b) |
| Galaxy 10 DECals | **95.42** | 94.86 | (Dagli, 2023) |
| iNAT-mini (val set) | **61.75** | NA | |

Table 4 shows the performance of WaveMix on image classification using supervised learning on ImageNet-1K on a single GPU with limited epochs. No augmentations were used to get a proper comparison of performance across the different architectures. WaveMix models outperform CNN and transformer-based models, and token-mixers. The use of non-learnable fixed weights and shallower network structure also makes inference using WaveMix faster than other architectures. See supplementary materials for inference throughput and GPU consumption.

Table 4: Results of Image classification on ImageNet-1K dataset ($224 \times 224$) using default Timm training script (Wightman, 2019) without augmentations on a single GPU shows improved accuracy and decreased parameter count by WaveMix.

| MODELS | # PARAMS | TOP-1 ACCURACY |
|---|---|---|
| ResNet-18 (He et al., 2015) | 11.7 M | 69.80% |
| ResNet-34 (He et al., 2015) | 21.8 M | 72.27% |
| MobileNetV3-large 0.75(Howard et al., 2019) | 4.0 M | 69.84% |
| ViT-base-patch-16 (Dosovitskiy et al., 2021) | 86.6 M | 66.93% |
| MLP-Mixer-base-patch-16 (Tolstikhin et al., 2021) | 59.9 M | 59.84% |
| PoolFormer-small-12 (Yu et al., 2021) | 11.9 M | 54.21% |
| ConvMixer-1024/20 (Trockman & Kolter, 2022) | 24.4 M | 74.57% |
| WaveMix-Lite-192/16 | 12.5 M | 70.82% |
| WaveMix-192/16 (level 3) | 27.9 M | **75.32%** |

Table 5: WaveMix needs very few parameters to achieve certain accuracy benchmarks compared to other architectures. Use of upsampling instead of upconvolution is denoted by *.

| DATASET | ACCURACY | MODEL | PARAMETERS |
|---|---|---|---|
| MNIST | 99% | WaveMix-Lite-8/10* | 3,566 |
| Fashion MNIST | 90% | WaveMix-Lite-8/5 | 7,156 |
| CIFAR-10 | 80% | WaveMix-Lite-32/7* | 37,058 |
| CIFAR-10 | 90% | WaveMix-Lite-64/6 | 520,106 |

### 4.5 Parameter efficiency

Table 5 shows that WaveMix meets certain accuracy benchmarks with far fewer parameters compared to previous models (Jha et al., 2021; Wu, 2018). Since WaveMix heavily uses kernels with fixed weights for token mixing, it needs far fewer parameters compared to commonly used architectures. We can further reduce the parameter count by replacing the upconvolution layers with upsampling layers using fixed interpolation techniques (e.g., inverse-DWT, bilinear, bicubic, pixel-shuffle).

### 4.6 Ablation studies

We performed ablation studies using ImageNet-1k and CIFAR-10 datasets on WaveMix to understand the effect of each type of layer on performance by removing the 2D-DWT layer, replacing it with Fourier transform or random filters, as well as learnable wavelets. All of these led to a decrease in accuracy. Those methods that did not reduce the feature map resolution led to an increase in GPU RAM consumption. When we removed the 2D-DWT layers from WaveMix, the GPU RAM requirement of the model increased by 62% and accuracy fell by 5%. This is due to the MLP receiving the full resolution instead of the half-resolution feature map from 2D-DWT. Replacing the 2D-DWT with the real part of a 2D-discrete Fourier transform (2D-DFT) showed 12% decrease in accuracy along with 73% increase in GPU consumption as the Fourier transform also does not downscale the feature map. Additionally, unlike the wavelet transform, the Fourier transform has spatially smoothly varying (not abrupt) kernels with global (as opposed to local) scope, which do not model object edges due to finite spatial extents in a sparse manner.

Replacing the filters of 2D-DWT with random filters of similar size as Haar wavelet resulted in 6% fall in accuracy, confirming that the fixed kernel weights of 2D-DWT is already well-suited for computer vision based on previous studies. GPU RAM consumption increased by 8% and accuracy decreased by 5% when we replaced the 2D-DWT with $2 \times 2$ max pooling, indicating that the loss of information by the latter hurts generalization.

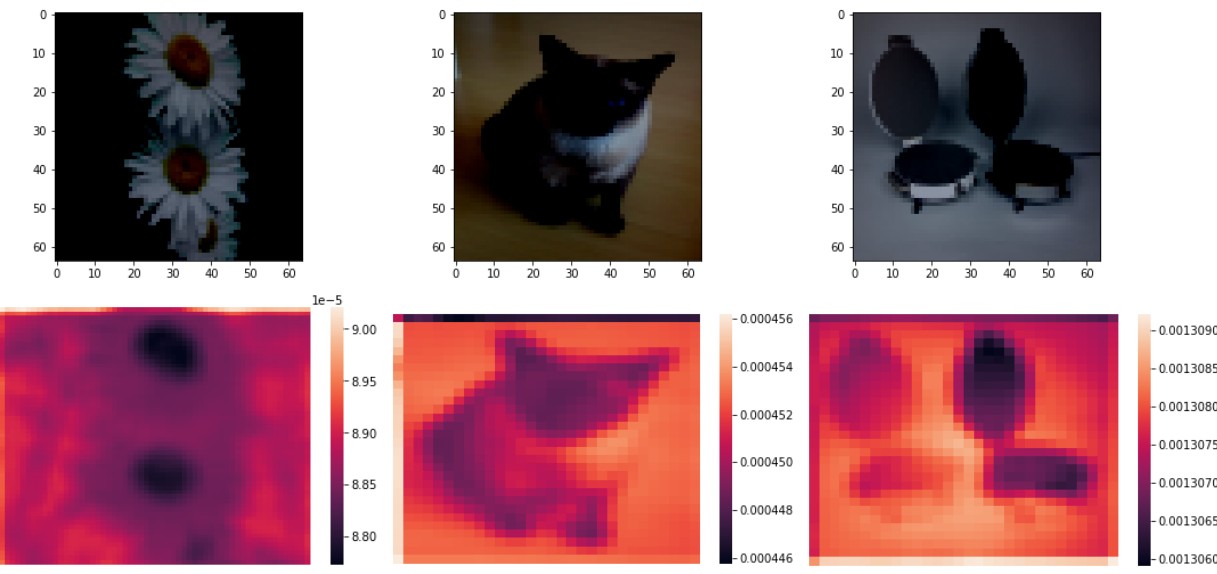

Figure 3: The results of occlusion analysis to find the significance of each pixel in the output decision shows that WaveMix identifies important regions (darker in second row) in an image for making the classification decision. The scale shows the probability of class output when the pixel is occluded.

We used lifting scheme (Bastidas Rodriguez et al., 2020) to create learnable wavelet coefficients and observed a decline in performance by 4% compared to fixed Haar wavelet coefficients. We also experimented with different families of wavelet coefficients, such as Daubechies, Coiflet, and Symlet series and observed that Haar wavelet was faster and gave more accurate test results than the others. We believe that although higher order Daubechies and other wavelets are perhaps better suited for information compression of natural images, the feature maps after the first convolutional layer are already sparse and are perhaps better analyzed by the Haar wavelet itself. The results of this experiment are provided in the Appendix.

We compared upsampling by transposed convolution in WaveMix block with (a) inverse-2D-DWT and (b) bi-linear upsampling layers. Both inverse 2D-DWT and bi-linear upsampling layers needed more WaveMix blocks compared to upconvolutions to give similar generalization. Even though the number of parameters of these individual layers was lower than upconvolution, the larger depth led to consumption of more GPU RAM and slower training and inference. Additional details of ablation studies on the number of blocks, levels of DWT, and embeddings and MLP dimension are included in the Appendix.

## 5 Conclusions, Future Directions, and Impact

We proposed a novel and versatile neural architectural framework – WaveMix – that can generalize at par with (and sometimes better than) both CNNs and vision transformers, and their hybrids for a variety of vision tasks, while needing fewer parameters, GPU RAM, and clock time. WaveMix uses multi-level 2D-DWT for lossless and image-appropriate down-sampling and token-mixing, that model image priors, such as scale-invariance, shift-invariance, and edge-sparseness. Unlike CNNs, WaveMix mixes tokens from far apart as it quickly expands the receptive field. Moreover, unlike transformers that unroll a 2D image into a 1D sequence, which makes them rigidly wedded to an image size or proportions, WaveMix handles the image in its 2D format itself, making it far easier to adapt it to various image sizes, proportions, and tasks.

Adapting WaveMix architecture to tasks, such as object detection and instance segmentation, would be worthwhile to explore, as would be scaling it to much deeper and wider proportions (increase $N$ and $C$). Additional ways of more comprehensive exploiting of multiple image priors must also be explored to reduce the resource requirements for training generalizable neural networks for vision.

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

# A    Appendix

## A.1    Feedforward Dimension and MLP Multiplication Factor

The feedforward dimension (ff) is the dimension of the embeddings of output from the MLP layer $m$ before it is passed to the deconvolution layer $t$. The deconvolution layer then changes the embedding dimension back to the value set in the model name. Unless otherwise mentioned, the value of feedforward dimension is set by default as the embedding dimension specified in the model name. Using a value higher than embedding dimension as ff dimension increases the number of parameters of the model and GPU consumption.

Feedforward dimension is different from the MLP multiplication factor (mul) which describes the increase in embedding dimension within the MLP layer (the first $1 \times 1$ convolution increases it by a factor and the second $1 \times 1$ convolution decreases it after is passes through the GELU activation). For example, MLP multiplication factor of 2 in a WaveMix-128 will use the first $1 \times 1$ convolutional layer inside MLP to increase the embedding dimension from 128 to 256. After the GELU activation, the second $1 \times 1$ convolutional layer inside MLP will decreases the embedding dimension back to 128. If we specify the ff dimension to be different from one provided in model name, then the second $1 \times 1$ convolutional layer inside MLP will change it to the ff dimension as specified. Unless otherwise mentioned, a MLP multiplication factor of 2 was used in all the models.

## A.2 Alternatives to Transposed Convolutions

We may also use parameter-free bi-linear up-sampling or pixel-shuffle operation for increasing feature resolution instead of the transposed-convolutions $t$ for further reduction in parameters and floating point operations (FLOPs). Since this can impact that learning capacity of the network, it is advised to add a convolutional layer after the parameter-free up-sampling operations whenever they are used.

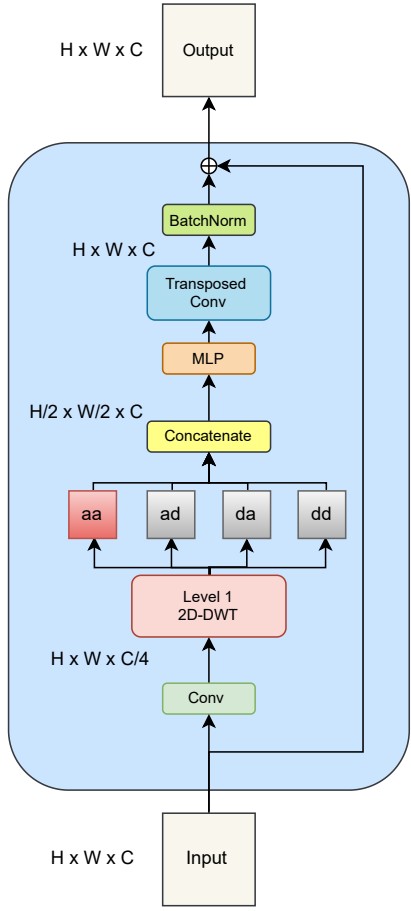

WaveMix-Lite Block (1-level 2D-DWT)

Figure 4: Details of the WaveMix-Lite block, which uses only a single level 2D-DWT

Table 6: Image classification performance on CIFAR-10 dataset by WaveMix-Lite-256/7 using different Wavelets on a T4 16 GB GPU show that Haar Wavelet is better. Training time in seconds/epoch is also reported.

| WAVELET | TOP-1 ACCURACY | S/EPOCH |
|---|---|---|
| **Haar (db1)** | **91.15%** | **25** |
| Daubechies (db2) | 90.72% | 44 |
| Daubechies (db3) | 89.14% | 68 |
| Daubechies (db4) | 88.00% | 104 |
| Daubechies (db5) | 86.88% | 148 |
| Biorthogonal (bior1.3) | 90.50% | 67 |
| Biorthogonal (bior1.5) | 89.61% | 157 |
| Reverse Biorthogonal | 90.14% | 67 |
| Coiflet (coif1) | 89.98% | 67 |
| Coiflet (coif2) | 85.68% | 233 |
| Symlet (sym2) | 90.24% | 44 |
| Symlet (sym3) | 88.87% | 63 |

### A.3 Semantic Segmentation

#### A.3.1 Detailed Results

The original input image resolution of $1024 \times 2048$ could not be send directly to WaveMix blocks in our experiments due to resource constraints. Two strided convolutional layers having stride of 2 each were used to reduce the input resolution to $256 \times 512$ before it reaches the WaveMix layers. Table 7 shows the detailed results of our experiments in Cityscapes dataset.

#### A.3.2 Ablation Studies

**Influence of input image size.** From 7 we can see that for the same model size, larger input image resolution gave better results. The results for $512 \times 1024$ input was 6-8% better than the corresponding results obtained while using input size of $256 \times 512$. Results for input size of $1024 \times 2048$ was 2 % better than $512 \times 1024$ input size.

**Influence of number of layers.** The number of layers that could be tested were limited due to the GPU constraints as well as the batch size requirements. We observed an increase in mIoU as the number of layers increases, then it peaks at around 16 layers for various input sizes and then gradually decreases for each additional layer.

**Influence of embedding dimension.** We varied the embedding dimension from 128 to 512. Variation of embedding dimension showed a behaviour similar to that shown by increasing the number of layers where mIoU first increases with increase in embedding dimension, then peaks at around 256, and then starts to decrease for both the input image sizes.

**Influence of the MLP multiplication factor.** Table 7 shows that increasing the multiplication factor (mul) does not increase the parameter count significantly. It can be used to vary the parameter count slightly for a marginal increase in performance. Increasing the MLP multiplication factor beyond 3 showed slight deterioration in performance with input images of size $256 \times 512$.

**Influence of multiple levels of Wavelet transforms** Table 7 shows that using multi-level 2D-DWT provides better results than just using a single level (WaveMix-Lite). The best results were obtained when using 4 levels of 2D-DWT for an input size of $1024 \times 2048$.

Table 7: Results for semantic segmentation using WaveMix models on Cityscapes validation dataset at different image resolutions in V100 and A100 GPUs without ImageNet-1k pre-training

| ARCHITECTURE | mIoU ↑ | # PARAM. ↓ | MAX BATCH FOR 16 GB ↑ | INFERENCE THROUGHPUT (FPS) ↑ |
|---|---|---|---|---|
| Image resolution 256 × 512 on 16 GB V100 GPU | | | | |
| WaveMix-Lite-128/8 | 63.33 | 2.9 M | 55 | 18 |
| WaveMix-Lite-128/20 (mul 3) | 67.76 | 7.5 M | 32 | 17 |
| WaveMix-Lite-160/12 | 65.08 | 6.6 M | 45 | 18 |
| WaveMix-Lite-192/12 | 65.92 | 9.5 M | 40 | 19 |
| WaveMix-Lite-224/12 | 66.67 | 12.9 M | 32 | 18 |
| WaveMix-Lite-256/7 (ff 160) | 65.30 | 7.2 M | 45 | 18 |
| WaveMix-Lite-256/7 (ff 160, mul 3) | 64.34 | 7.9 M | 40 | 19 |
| WaveMix-Lite-256/7 (ff 192, mul 3) | 65.27 | 8.9 M | 30 | 17 |
| WaveMix-Lite-256/12 (ff 160) | 67.11 | 11.5 M | 30 | 18 |
| WaveMix-Lite-256/12 (ff 192) | 65.46 | 13.3 M | 25 | 17 |
| WaveMix-Lite-256/12 (ff 224) | 66.75 | 15.0 M | 30 | 18 |
| WaveMix-Lite-256/12 | 67.46 | 16.9 M | 25 | 12 |
| WaveMix-Lite-256/12 (ff 1024, mul 3) | 62.39 | 63.2 M | 22 | 17 |
| WaveMix-Lite-256/16 (ff 272, mul 3) | 67.46 | 25.5 M | 20 | 17 |
| **WaveMix-Lite-256/16 (ff 512, mul 3)** | **71.75** | 44.2 M | 18 | 18 |
| WaveMix-Lite-256/16 (ff 512, mul 4) | 67.17 | 47.3 M | 18 | 18 |
| WaveMix-Lite-256/16 (ff 1024, mul 3) | 69.94 | 84.0 M | 18 | 17 |
| WaveMix-Lite-256/18 (ff 512, mul 3) | 65.16 | 49.6 M | 16 | 17 |
| WaveMix-Lite-256/20 (mul 3) | 67.65 | 30.1 M | 16 | 18 |
| WaveMix-Lite-256/20 (ff 512, mul 3) | 67.80 | 55.0 M | 16 | 17 |
| WaveMix-Lite-272/16 | 67.48 | 25.0 M | 22 | 17 |
| WaveMix-Lite-288/16 | 67.90 | 28.0 M | 20 | 17 |
| WaveMix-Lite-304/16 | 67.76 | 31.2 M | 18 | 18 |
| WaveMix-Lite-320/16 (ff 512, mul 3) | 67.94 | 56.5 M | 15 | 16 |
| WaveMix-Lite-512/12 (ff 1024, mul 3) | 65.09 | 133.2 M | 11 | 17 |
| Image resolution 512 × 1024 on 40 GB A100 GPU | | | | |
| WaveMix-Lite-128/8 | 67.55 | 2.9 M | 32 | 16 |
| WaveMix-Lite-256/7 | 70.43 | 10.2 M | 18 | 16 |
| WaveMix-Lite-256/14 | 73.86 | 19.5 M | 15 | 16 |
| **WaveMix-Lite-256/16** | **76.79** | 22.2 M | 13 | 16 |
| WaveMix-Lite-256/17 | 74.26 | 23.5 M | 12 | 16 |
| WaveMix-Lite-256/18 | 74.67 | 24.9 M | 12 | 16 |
| WaveMix-Lite-272/16 | 73.21 | 25.1 M | 12 | 16 |
| WaveMix-Lite-288/16 | 73.06 | 28.1 M | 11 | 16 |
| Image resolution 1024 × 2048 on 80 GB A100 GPU | | | | |
| WaveMix-Lite-256/16 | 75.32 | 22.3 M | 1 | 9 |
| WaveMix-256/16 (level 2) | 75.92 | 35.9 M | 1 | 8 |
| WaveMix-256/16 (level 3) | 76.54 | 49.6 M | 1 | 7 |
| WaveMix-256/15 (level 4) | 77.18 | 59.4 M | 1 | 7 |
| **WaveMix-256/16 (4 level)** | **78.64** | 63.2 M | 1 | 7 |
| WaveMix-256/17 (level 4) | 77.68 | 67.1 M | 1 | 6 |

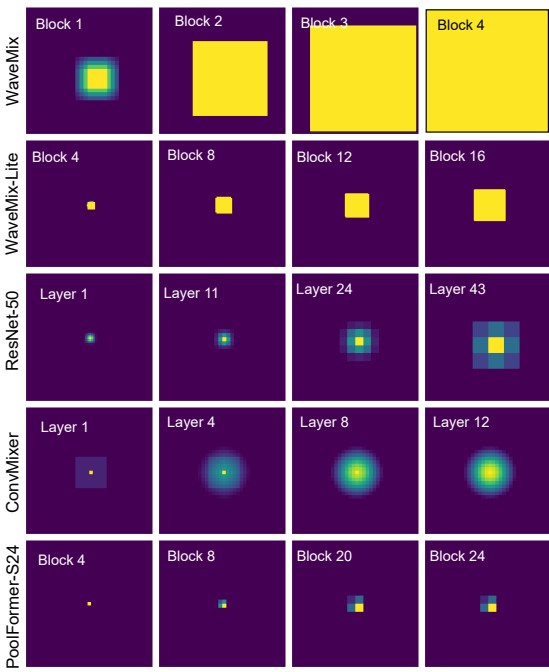

Figure 5: Visualisation of receptive fields for different models show a rapid expansion of receptive field in WaveMix as we add layers or blocks due to multi-level 2D-DWT. A blank image with a single high pixel value near the center was sent as input to the models. All parameters were assigned a value of one and all bias were set to zero.

Table 8: Performance of WaveMix on different datasets. SOTA results are highlighted in bold. *Architecures pre-trained on ImageNet-1k. †Used TrivialAugment Müller & Hutter (2021).

| DATASETS | MODEL | INPUT RESOLUTION | ACC. (%) |
|---|---|---|---|
| By_Class | WaveMix-Lite-128/7 | $28 \times 28$ | **88.43** |
| Balanced | WaveMix-Lite-128/7 | $28 \times 28$ | **91.06** |
| Letters | WaveMix-Lite-112/16 | $28 \times 28$ | **95.96** |
| Digits | WaveMix-Lite-112/16 | $28 \times 28$ | **99.82** |
| By_Merge | WaveMix-Lite-128/16 | $28 \times 28$ | **91.80** |
| Places-365 | WaveMix-240/12 (level 4) | $224 \times 224$ | **56.45** |
| iNat-2021 | WaveMix-256/16 (level 2) | $256 \times 256$ | **61.75** |
| SVHN | WaveMix-Lite-144/15 | $32 \times 32$ | 98.73 |
| CIFAR-10 † | WaveMix-Lite-320/8 | $32 \times 32$ | 95.37 |
| CIFAR-100 † | WaveMix-Lite-256/10 | $32 \times 32$ | 78.78 |
| Tiny ImageNet † | WaveMix-244/12 (level 2) | $64 \times 64$ | 64.51 |
| CIFAR-10 † | WaveMix-192/16 (level 3)* | $192 \times 192$ | 97.61 |
| CIFAR-100 † | WaveMix-192/16 (level 3)* | $224 \times 224$ | 85.64 |
| SVHN | WaveMix-192/16 (level 3)* | $128 \times 128$ | 98.79 |
| Tiny ImageNet † | WaveMix-192/16 (level 3)* | $128 \times 128$ | 77.49 |

Table 9: Image classification results on ImageNet-1K dataset (224×224) without augmentations by WaveMix (arrows in column headers show desired directions). Patch size of 5 is used for WaveMix-Lite models and 4 is used for other WaveMix models. Training was done using Timm training script Wightman (2019).

| MODELS | TOP-1 ACCU. (%) ↑ | # PARAM. ↓ |
|---|---|---|
| WaveMix-Lite-192/16 | 70.82 | 12.5 M |
| WaveMix-Lite-256/16 | 71.45 | 23.1 M |
| WaveMix-192/16 (2 level) | 72.30 | 20.5 M |
| WaveMix-256/8 (3 level) | 72.72 | 26.2 M |
| **WaveMix-192/16 (3 level)** | **75.32** | 27.9 M |

Table 10: Image classification on ImageNet-1K dataset (224 × 224) without augmentations shows improved accuracy as well as throughput due to decreased parameter count and GPU RAM consumption by WaveMix (arrows in column headers show desired directions). Timm training script Wightman (2019) is not used.

| ARCHITECTURE | TOP-1 ACCU. (%) ↑ | # PARAM. ↓ | GPU RAM fOR BATCH SIZE 64 ↓ | THROUGHPUT (IM/S) TRAIN ↑ | TEST ↑ |
|---|---|---|---|---|---|
| ResNet-18 | 55.12 | 11.7 M | 2.7 GB | 450 | 439 |
| ResNet-34 | 57.02 | 21.8 M | 3.1 GB | 414 | 410 |
| ResNet-50 | 61.76 | 25.6 M | 6.2 GB | 638 | 617 |
| ResNet-101 | 64.60 | 44.5 M | 9.6 GB | 487 | 725 |
| ResNet-152 | 65.86 | 60.2 M | 12.7 GB | 344 | 758 |
| MobileNetv3-small | 51.57 | 2.5 M | 1.4 GB | 255 | 229 |
| MobileNetv3-large | 58.89 | 5.5 M | 3.5 GB | 492 | 481 |
| ViT-B-16 | 39.53 | 86.6 M | 10.0 GB | 140 | 420 |
| ViT-B-32 | 30.11 | 88.2 M | 2.2 GB | 1,595 | 1,613 |
| ConvMixer-512/12 | 60.24 | 4.2 M | 10.8 GB | 292 | 735 |
| ConvMixer-512/16 | 62.24 | 5.4 M | 14.1 GB | 220 | 725 |
| ConvMixer-1024/12 | 64.13 | 14.6 M | 23.6 GB | 251 | 667 |
| WaveMix-Lite-128/8 | 54.12 | 3.9 M | 4.5 GB | 1,242 | 1,724 |
| WaveMix-224/12 (2 level) | 65.90 | 22.6 M | 9.8 GB | 385 | 1,250 |
| **WaveMix-240/12 (3 level)** | **70.02** | 33.8 M | 15.6 GB | 205 | 610 |

## A.4 Image Classification

### A.4.1 Results on Smaller Resolution Image Datasets

We observe from Table 9 that deeper WaveMix models perform better and this suggests that even further scale-up of WaveMix in multi-GPU setting could be possible. We see from Table 10 that shallow WaveMix models are competitive in performance to deeper ResNets and MobileNets, and they are approximately 2x faster in training 4x faster in inference. Even deeper WaveMix models provide faster inference and better performance compared to the other models. ConvMixers are parameter-efficient and provide high accuracy, but they need much higher GPU RAM compared to WaveMix.

In Table 11 we see that on CIFAR and TinyImageNet datasets, WaveMix-Lite performs much better than the other models, giving accuracy higher than ResNets and MobileNets with 4 to 10 times fewer parameters and less GPU consumption. GPU consumption of WaveMix-Lite is sometimes 50 times lower for similar performance when compared to transformer models.

Table 11: Results for image classification on small datasets (32x32, 64x64) show improved accuracy as well as decreased parameter count and GPU RAM consumption by WaveMix-Lite. No augmentations were used.

| MODEL | #PARAM. ↓ | GPU RAM FOR BATCH SIZE 64 ↓ | ACC. (%) ↑ CIFAR-10 | CIFAR-100 | TINYIMGNT |
|---|---|---|---|---|---|
| ResNet-18 Hassani et al. (2021) | 11.20 M | 1.2 GB | 90.27 | 63.41 | 48.11 |
| ResNet-34 Hassani et al. (2021) | 21.30 M | 1.4 GB | 90.51 | 64.52 | 45.60 |
| ResNet-50 Hassani et al. (2021) | 25.20 M | 3.3 GB | 90.60 | 61.68 | 48.77 |
| MobileNetV2 Hassani et al. (2021) | 8.72 M | - | 91.02 | 67.44 | - |
| ViT-128/4×4 | 0.53 M | 13.8 GB | 56.81 | 30.25 | 26.43 |
| ViT-384/12x6 Hassani et al. (2021) | 85.60 M | - | 76.42 | 46.61 | - |
| ViT-Lite-256/6x4 Hassani et al. (2021) | 3.19 M | - | 90.94 | 69.20 | - |
| HybridViN-128/4×4 | 0.62 M | 4.8 GB | 75.26 | 51.44 | 34.05 |
| CCT-128/4×4 | 0.91 M | 15.8 GB | 82.23 | 57.09 | 39.05 |
| CvT-128/4×4 | 1.12 M | 15.4 GB | 79.93 | 48.29 | 40.69 |
| MLP-Mixer-512/8 | 2.41 M | 1.0 GB | 72.22 | 44.23 | 26.83 |
| WaveMix-Lite-16/7 | 0.04 M | 0.1 GB | 64.98 | 23.03 | 19.15 |
| WaveMix-Lite-32/7 | 0.15 M | 0.3 GB | 84.67 | 46.89 | 34.34 |
| WaveMix-Lite-64/7 | 0.60 M | 0.6 GB | 87.81 | 62.72 | 46.31 |
| WaveMix-Lite-128/7 | 2.42 M | 1.1 GB | 91.08 | 68.40 | 52.03 |
| WaveMix-Lite-144/7 | 3.01 M | 1.2 GB | **92.97** | 68.86 | 52.38 |
| Wavemix-Lite-160/13 | 6.90 M | 9.4 GB | - | - | **54.76** |
| WaveMix-Lite-256/7 | 9.62 M | 2.3 GB | 90.72 | **70.20** | 51.37 |

Table 12: Comparison of Top-1 Accuracy of WaveMix models using different initial convolution layers on ImageNet-1k dataset. Patch size of 4 is used in patchify layer

| MODEL | STRIDED CONVOLUTIONS | PATCHIFYING LAYER |
|---|---|---|
| WaveMix-Lite-224/20 | 62.79 | 67.91 |
| WaveMix-240/12 (2 level) | 64.75 | 66.44 |
| WaveMix-240/12 (3 level) | 67.97 | 70.02 |

### A.4.2 Augmentations and Learning Rate Tuning

The previously reported results for the other architectures include the effect of various well-intentioned incremental training methods (tips and tricks) like Timm training script Wightman (2019), including RandAugment Cubuk et al. (2019), mixup Zhang et al. (2017), CutMix Yun et al. (2019), random erasing Zhong et al. (2017), gradient norm clipping Zhang et al. (2020), learning rate warmup Gotmare et al. (2019) and cooldown. These additional methods improve the results of the core architectures trained using simple methods by a few percentage points each. For example, Mixup, Cutmix, Random Erasing, RandAugment, Random Scaling and Gradient Norm Clipping improved accuracy of ConvMixer by 9.55 percentage points in image classification Trockman & Kolter (2022). However, experimenting with these additional training methods requires extensive hyperparameter tuning. On the other hand, by excluding these methods, we were able to compare the contribution of the base architectures in a uniform manner. The accuracy obtained in our experiments for the other architectures are thus lower than the previously reported numbers, but the results are still within the expected range when such training methods are not used. We have used TrivialAugment Müller & Hutter (2021) while training some of the smaller datasets and observed a higher performance compared to RandAugment Cubuk et al. (2019). Some results using TrivialAugment as the only data augmentation method used for training is reported in Table 8.

captionVariation of performance and resource consumption of WaveMix-Lite-144/7 on classification of CIFAR-10 dataset using different levels of 2D-DWT separately.

| LEVEL OF 2D-DWT | ACCU. (%) | # PARAM. | GPU FOR BATCH SIZE OF 1024 |
|---|---|---|---|
| 1 | **91.61** | 3 M | 19.6 GB |
| 2 | 87.39 | 5.9 M | 14.2 GB |
| 3 | 78.07 | 15.2 M | 13.7 GB |
| 4 | 65.40 | 47.7 M | 13.2 GB |

### A.4.3  Ablation Studies

**Influence of initial convolutional layers.**  We experimented with two types of convolutional layers in the initial convolutional layer to downscale the image size for large resolution datasets. Strided convolutions with stride of 2 reduced the resolution by half in each layer. In patchifying convolutional layer, we used 2 layers of normal convolutions followed by a patchifying convolutional layer (stride and kernel size set to same value), a GELU non-linearity and a batch normalisation layer. We see from Table 12 that using patchifying convolutions perform better than strided convolutions for WaveMix models.

**Influence of levels of 2D-DWT.**  Table A.4.3 shows the variation of performance and resource consumption for each level of 2D-DWT. Level-1 2D-DWT reduced image resolution by half, reducing the computational cost and GPU consumption compared to convolutional layers. Using higher levels of 2D-DWT could further reduce the image resolution to one-fourth, one-eight and so on which can further reduce the computational costs. But the deconvolution layer used to resize the output back to input size will need a lot more parameters. This will consume more resources in terms of GPU and time. Each increment in the level of 2D-DWT results in doubling of the number of parameters, but provides only a very small reduction in GPU consumption, especially when we go to higher levels of 2D-DWT. In each of the higher level decomposition, when the approximation and detail coefficients are concatenated as a tensor, the noise intensity in the detail coefficients would be stronger than that of useful details (object edge, texture, or contour, etc.) which could be the cause of degradation in the performance. When we use multiple levels of 2D-DWT together, we observe a gain in performance as we add more levels as shown in Table 9 for high resolution images. This improvement in performance was not observed for low resolution images when we used more than 1 level of 2D-DWT (WaveMix-Lite).

**Influence of number of layers.**  The performance of WaveMix models generally improve as the number of layer increases. The behaviour observed in smaller datasets show that the accuracy increases with increase in number of layers, peaks at a particular value and then do not show any increase for any further addition of layers. In experiments with ImageNet-1k dataset, we observed that while using higher levels of 2D-DWT, the number of layers needed is much lower for reaching the best performance when compared to WaveMix-Lite. Therefore, WaveMix with multi-level 2D-DWT are shallower than WaveMix-Lite.

**Influence of the Embedding Dimension.**  Our experiments showed that increasing the embedding dimension of a model usually improved the model performance, but the resource-utilization also increased significantly. Doubling the embedding dimension of model from 128 to 256 results in an increase of parameter count by more than three times and doubles the GPU RAM consumption.

