# OpenReview forum: "WaveMix: Resource-efficient Neural Network for Image Analysis"
_TMLR — Rejected by TMLR_

### Review · Reviewer_erki · 2024-11-06

**Summary Of Contributions:**

The paper introduces WaveMix, a new block for building resource-efficient neural networks capable of solving classification and segmentation tasks. WaveMix uses 2D discrete wavelet transforms (2D-DWT) to decompose images and feature maps into their frequency components. This decomposition allows WaveMix to exploit image priors such as scale invariance, shift invariance, and edge sparsity.

**Audience:**

Yes

**Claims And Evidence:**

No

**Requested Changes:**

From my point of view, this work requires a more thorough review of prior research, as many relevant works appear to be overlooked. In particular, there is no comparison with [1], where a similar idea seems to have been proposed in a more straightforward, drop-in replacement form. Additionally, the paper lacks comparison with true state-of-the-art methods, suggesting that recent advancements from the past two years were missed. The baselines presented also appear weak, and some claims made in the paper lack supporting evidence or detailed reporting. Given these issues, I am inclined to recommend rejection due to the insufficient comparison with recent methods, incomplete reporting, and inadequate baselines.

**Strengths And Weaknesses:**

Strengths:

The idea of using 2D discrete wavelet transforms to create more efficient neural networks is intriguing, and the use of simple MLPs instead of convolutions on the wavelet coefficients is innovative. In this approach, only channel mixing and frequency component mixing are applied, while spatial mixing is omitted. Spatial mixing occurs later through a transposed convolution applied after the MLP.

Weaknesses:

-	A comparison to prior work [1] is missing. Most importantly, there is no comparison to Wavelet Convolutions, which also leverage wavelet decomposition to increase receptive context. In that approach, the convolutional block can be directly replaced without requiring a new architectural block, making it more comparable to other existing methods.
-	Comparisons to other works that reduce the number of learnable parameters for image classification and segmentation, such as [2,3], are missing.
-	In Table 3, prior work on Places-356 and iNat datasets performs significantly better than what is reported here (e.g., [4,5]).
-	Table 4 only reports ResNets up to size 34, which has 21.8M parameters. If ResNet-50, with 25.6M parameters and a reported accuracy of 76.13% (PyTorch), were included, it would easily outperform the WaveMix model, which has 27.9M parameters but only achieves 75.32% accuracy.
-	The supplementary materials promised a GPU RAM evaluation for the iNat dataset, but this information was not found in the main paper.
-	The model’s training and inference runtime are not reported, despite claims in the abstract of faster forward and backward passes as well as inference.
-	The authors claim faster clock times in the conclusion but do not provide any comparisons with other approaches.
-	Table 5 is supposed to include a comparison to prior work [6,7], but only WaveMix models are reported.
-	The evaluation in Table 11 appears to rely on weak baselines; for instance, CIFAR-10 accuracies of 90% for ResNet-18 are considerably lower than baseline values reported in [2], for example.
-	The qualitative assessment in Figure 2 should include at least one other approach to enable a better comparison of the proposed method against state-of-the-art techniques.
Questions:
-	Why is a transposed convolution used to combine the wavelet coefficients? Would it not be more efficient to perform an inverse DWT on the four wavelet components? This approach could also incorporate lower-level wavelet decompositions directly into the "low-low" component of the wavelet decomposition.
-	The computational cost of calculating the DWT is not reported. How much does this impact runtime?
-	Why are augmentations not used? This is standard practice for state-of-the-art results.

Minor

-	The first sentence in Section 3.2 needs to be revised for clarity.

[1] Finder, Shahaf E., et al. "Wavelet Convolutions for Large Receptive Fields." European Conference on Computer Vision. Springer Nature, 2024.

[2] Grabinski, Julia, et al.  "As large as it gets-Studying Infinitely Large Convolutions via Neural Implicit Frequency Filters." Transactions on Machine Learning Research, 2024.

[3] Wang, Wenhai, et al. "Internimage: Exploring large-scale vision foundation models with deformable convolutions." Proceedings of the IEEE/CVF conference on computer vision and pattern recognition. 2023.

[4] He, Kaiming, et al. "Masked autoencoders are scalable vision learners." Proceedings of the IEEE/CVF conference on computer vision and pattern recognition. 2022.

[5] Ryali, Chaitanya, et al. "Hiera: A hierarchical vision transformer without the bells-and-whistles." International Conference on Machine Learning. PMLR, 2023.

[6] Debesh Jha, Anis Yazidi, Michael A. Riegler, Dag Johansen, Håvard D. Johansen, and Pål Halvorsen. Lightlayers: Parameter efficient dense and convolutional layers for image classification, 2021. URL https: //arxiv.org/abs/2101.02268.

[7] Chai Wah Wu. Prodsumnet: reducing model parameters in deep neural networks via product-of-sums matrix decompositions, 2018. URL https://arxiv.org/abs/1809.02209.

---

### Review · Reviewer_nupf · 2024-11-08

**Summary Of Contributions:**

This paper propose a generalizable, scalable, and efficient architecture for image analysis tasks, named as WaveMix network. The core building block that leads to the optimal efficiency is the multi-level two-dimensional discrete wavelet transform (2D-DWT) operation.  2D-DWT enables efficient token-mixing and can increase receptive field exponentially. In addition, low-frequency and high-frequency components are equally focused in the design of WaveMix block. The proposed architecture are extensively verified on segmentation and classification tasks, demonstrating superior efficiency and effectiveness.

**Audience:**

Yes

**Claims And Evidence:**

No

**Requested Changes:**

See weakness above

**Strengths And Weaknesses:**

Strengths
- The presentation quality is good, papers are well-written and easy to follow.
- Comprehensive experiments cover both segmentation and classification tasks.
- Impressive efficiency and performance compared to existing models for vision tasks (CNN/ViTs).

Weakness
- Some baseline results in experiments are doubtful. For example, 66.93\% for ViT-B-16 is too low and it takes very few epochs for it to be trained with top-1 acc >70%. I understand that author claim they train on a single GPU with limited epochs, but this unfair comparison do not convince me.
- Efficient architecture, such as MobileNets, EfficientNet, should also be discussed and compared.
- Is scaling law for transformer-based model applicable to the proposed WaveMix? Can WaveMix be generalized to generation tasks?

---

### Review · Reviewer_PpTi · 2024-11-18

**Summary Of Contributions:**

This paper studies the field of computer vision, specifically tasks like classification and semantic segmentation. The paper argues that prior works, like CNN, ViT, and token mixers, do not exploit image priors like scale-invariance and spatial sparseness of edges. Therefore, the author proposes WaveMix, which models information from three priors, namely, scale-invariance, shift-invariance, and sparseness of edges. The WaveMix architecutre compared to previous works (CNN, ViT and token mixers) is also less computationally expensive.

**Audience:**

Yes

**Claims And Evidence:**

Yes

**Requested Changes:**

Please see the weaknesses above.

**Strengths And Weaknesses:**

# Strengths
1.	The work is very interesting since it proposes an alternative architecture to CNN, ViT, and token mixers.
2.	The proposed architecture is computationally efficient and shows good results.
3.	The paper is well written.

# Weaknesses

## Major Comments

1. The abstract of the paper starts as “We propose a novel neural architecture for computer vision.” This immediately tells me, as a reviewer, that the proposed work caters to the whole computer vision field and will be demonstrated in the paper. Further, in the last sentence of the Abstract and the First row of Table 1 in the Introduction, the author mentions that one of the tasks performed is segmentation. Whereas the actual task performed is semantic segmentation. Therefore, I think that the claim in the Abstract and Introduction is somewhat misleading. The authors should limit the scope to what they have experimented upon.
2. One of the important properties of Wavemix block is that it performs spatial resolution contraction followed by expansion back to the original size. This immediately warrants at least a discussion, if not a comparison with AutoEncoders. I think the readers of this paper would benefit from this.
3. The comparison results do not report any statistical significance testing or standard deviation. The authors should include them.
4. The most latest baseline the model compares with on semantic segmentation and ImageNet-1K classification are [1] and [2], respectively, and both of them are from 2022. There are many new baselines that the author should compare their model with. I am pointing out one for semantic segmentation [3]. There are other baselines, too. Simply following the papers citing [1] gives recent baselines. The same applies to image classification on ImageNet-1K and other datasets. I encourage the authors to include appropriate and recent baselines for both tasks.

[1] Zhou, Daquan, et al. "Understanding the robustness in vision transformers." International Conference on Machine Learning. PMLR, 2022.

[2] Trockman, Asher, and J. Zico Kolter. "Patches Are All You Need?." Transactions on Machine Learning Research.

[3] Fan, Qihang, et al. "Rmt: Retentive networks meet vision transformers." Proceedings of the IEEE/CVF Conference on Computer Vision and Pattern Recognition. 2024.

## Minor Comments
1. Typo in section 4.3: “table 1”
2. Reference citations in the paper are not coherent. For many accepted papers, the Arxiv version is cited. Please correct them and, wherever applicable, provide the citations with the venue where the papers are accepted.

---

### Decision · Action_Editor_ubwj · 2024-12-19

**Recommendation:** Reject

**Comment:**

The authors did not respond to the reviewers' questions and concerns during discussion phase. All three reviewers recommend rejection.

**Audience:**

Yes, people interested in computer vision could find interest also in the findings of this paper

**Claims And Evidence:**

The reviewers raised a number of questions and concerns regarding missing related works and non-convining experimental results. These remained unanswered from the side of the authors.